# Differences in Driving Intention Transitions Caused by Driver’s Emotion Evolutions

**DOI:** 10.3390/ijerph17196962

**Published:** 2020-09-23

**Authors:** Yaqi Liu, Xiaoyuan Wang

**Affiliations:** 1School of Transportation and Vehicle Engineering, Shandong University of Technology, Zibo 255000, China; liuyaqi518@126.com; 2College of Electromechanical Engineering, Qingdao University of Science & Technology, Qingdao 266000, China; 3Joint Laboratory for Internet of Vehicles, Ministry of Education-China Mobile Communications Corporation, Tsinghua University, Beijing 100048, China

**Keywords:** driving intention, transition difference, emotion, HMM

## Abstract

Joining worldwide efforts to understand the relationship between driving emotion and behavior, the current study aimed at examining the influence of emotions on driving intention transition. In Study 1, taking a car-following scene as an example, we designed the driving experiments to obtain the driving data in drivers’ natural states, and a driving intention prediction model was constructed based on the HMM. Then, we analyzed the probability distribution and transition probability of driving intentions. In Study 2, we designed a series of emotion-induction experiments for eight typical driving emotions, and the drivers with induced emotion participated in the driving experiments similar to Study 1. Then, we obtained the driving data of the drivers in eight typical emotional states, and the driving intention prediction models adapted to the driver’s different emotional states were constructed based on the HMM severally. Finally, we analyzed the probabilistic differences of driving intention in divers’ natural states and different emotional states, and the findings showed the changing law of driving intention probability distribution and transfer probability caused by emotion evolution. The findings of this study can promote the development of driving behavior prediction technology and an active safety early warning system.

## 1. Introduction

Traffic safety continues to come into focus with the ever-increasing use of cars in modern society, and traffic accidents are becoming a major factor causing human injuries [1]. Although the human, social and economic cost of traffic accidents is largely preventable, there has been insufficient action to combat this global challenge [2]. It has been proven that human factors predict a greater amount of variance in road accidents than vehicle and road factors do [3]. More concretely, research showed that 90% of rear-end collisions and 60% of frontal collisions would be avoided if the driver realized the danger and took effective measures one second in advance [4]. However, in car driving activities, human drivers have inherent limitations in perception, decision-making, and behavior execution. Even experienced drivers may ignore important information in the driving environment and make wrong judgments about traffic safety situation, and then adopt inappropriate or unsafe driving behaviors [5]. Therefore, it is very important to monitor and predict driving behavior through the on-board intelligence system, and to evaluate whether the driving behavior can keep the car in a safe state in a specific environment [6]. Many researchers in the field of automotive active safety have begun to build driving behavior identification and prediction models [7,8,9]. However, due to the complexity of the human behavior mechanism, there is still a lack of driving behavior prediction algorithms that can be applied to vehicle intelligence systems. In recent years, researchers argued that driver’s affective factors have close associations with road accidents [10,11]. Psychologists now widely accept that it is impossible for people to think or perform an action without engaging their emotional system, at least unconsciously [12,13]. Considering the previous driving behavior prediction models often ignore emotion that is a key factor affects driving behavior, the practical studies on the influence of emotion on driving behavior and related applications for driving behavior prediction are urgently needed [14,15]. The purpose of this study was to reveal the differences in driving intention transitions caused by driver’s emotion evolutions—i.e., to study the generation and transition differences in driving intentions when a is driver faced with the same or similar driving environments but in different emotional states. The intention is the psychological variable that has the closest relationship with real behavior [16] and is a measurement index, with high accuracy, for the human behavior prediction [17]. We expected that the research results could provide theoretical support for the construction of driving behavior prediction models that consider emotional factors.

In this study, the driving intention prediction model for a driver’s natural state, and eight typical driving emotion states, based on the Hidden Markov Model (HMM), were established and trained. In the intention prediction model, we used a hidden Markov chain to describe the random process of a driver’s intentions changing with time, and took the vehicle driving data corresponding to the driver’s different intention transitions as the observed value of the hidden Markov chain. On the premise of model accuracy verification, we extracted the probability distribution of different driving intentions, the transition probability matrix between different driving intentions, and the probability distribution of driving intentions corresponding to different observation values from the HMMs. For the driver’s natural state and different emotional states, we obtained the overall differences in driving intention probability distribution and transition probability under different emotional states through the comparative analysis. For the probability distribution of driving intention corresponding to different observed values, the Bayesian formula was used to transform it into the probability distribution of driving intention under different driving environments (observation values)—i.e., the probability distribution of driving intention generated by drivers under different environmental states. Finally, we compared and analyzed the probability distribution of driving intention under the same driving environment parameters but different emotional states and obtained the driving intention transition law caused by emotional evolution.

## 2. Study 1-A: Driving Intention Prediction Model Based on HMM

### 2.1. Materials and Methods

#### 2.1.1. Experiment Design

Participants

Sixty-two drivers (33 males and 29 females) aged from 20 to 48 (M = 28.61, SD = 6.83) participated in this study. The participants recruited were undergraduate students, urban residents and taxi drivers in Zibo City, China. All of the participants were licensed drivers and their driving experience ranged from 1 to 16 years (M = 4.47, SD = 2.98).

Measurement of Driver’s Expected Speed and Distance

The expected speed and expected distance are important parameters in driving behavior studies [18]. Expected speed means the maximum safe driving speed that the driver wants to achieve when the vehicle is running free from constraints of other vehicles [19]. Expected distance is a concept in the car-following theory [20,21,22,23]. The expected distance mentioned in this study was the distance from the head of the following car to the rear of the leading car in the car-following scene. We used the virtual driving platform (Figure 1b) to measure the participants’ expected speed and distance. We constructed a free driving scene to test the participants’ expected speed. In this scene, traffic flow and intersection signals were not set, and the speed limit was 60 km/h. We asked the participants to drive at their own speed on the virtual road under the condition of obeying traffic rules and ensuring driving safety. The driving time was 5 min, and we took the average driving speed of the middle 3 min as the expected speed of each participant. Besides, we constructed a car-following scene to test the participants’ expected distance. The traffic flow and intersection signals were also not set. In this scenario, a virtual vehicle was set to drive at a uniform speed at 60 km/h. Participants were required to follow the vehicle in the virtual environment. Under the condition of obeying basic traffic rules and ensuring driving safety, the participants can freely choose the driving speed and following distance. The driving was 5 min, and we took the average distance between the two cars of the middle 3 min as the expected distance of each participant.

Driving Experiment

The driving experiments included actual driving and virtual driving experiment. We used two experiment vehicles equipped with a multi-function speed-measuring instrument, laser range sensor, video capture system, and notebook computer for the actual driving experiments (Figure 1a), and used a virtual driving platform for the virtual driving experiment (Figure 1b). In the actual driving experiment, we used two vehicles to simulate the car-following scene. One of the vehicles, as the front vehicle, ran normally along the experiment route with a velocity of 60 ± 5 km/h and a participant drove the following one. The urban roads, which are full of intersections (signalized and no signalized) and corners, were not the ideal experimental environment. Considering the high safety risks involved in driving experiments, the enclosed expressway was not selected either. Finally, we chose a semi-enclosed road section of Zhangbo Road (Figure 2a) that was located far out of the city and connected the Zhangdian District and Boshan District as the actual driving experiment route. The whole length of the route was about 9.8 km (between points A and B in Figure 2a) and had a 70 km/h speed limit. The experiment vehicles were essentially independent of the crossing or merging effects of the vehicles on other roads except for a signalized intersection (due to regional conditions, the probability of signalized intersection was minimized), which was noted as point C in Figure 2a. In order to avoid the mandatory effect of the signalized intersection on driving intention, we did not use the data collected from 100 m before and after the intersection. We organized the actual driving experiments in off-peak periods under good weather and road conditions. We applied the virtual driving platform (Figure 1b) to carry out the virtual driving experiment. The virtual road section was structured (Figure 2b) in the platform according to the road basic properties and traffic volume of the actual experiment route. We created the car-following scene and collected the driving data in the platform.

Data Pre-Processing

Each participant took part in a 10-min actual driving experiment and a 10-min virtual driving experiment. Table 1 shows the data obtained through the experiment. To describe the relationship between vehicle state and driver’s psychological expectation, $v_{1}$ and $v_{e}$ were coded as $\Delta v_{e}=v_{1}-v_{e}$, and $\Delta v_{e}$ was defined as expected speed deviator. d and $d_{e}$ were coded as $\Delta d_{e}=d-d_{e}$, and $\Delta d_{e}$ was defined as the expected distance deviator. We used the acceleration to represent the execution result of intention. The time window for a single state was set to 10 s (the F-test was used to compare the model parameters at a single time window of 5, 10 and 15 s, respectively, and no significant difference was found. The test results are shown in Appendix A). Table 2 shows the rules for the data discretization. Finally, we obtained 3039 and 3534 data sets from the actual and virtual driving experiments, respectively.

#### 2.1.2. Model Construction

Hidden Markov Model (HMM)

The HMM is a statistical analysis model that can be represented a set of hidden states $Q=\{q_{1},q_{2},\ldots ,q_{N}\}$, a set of observable states $O=\{o_{1},o_{2},\ldots ,o_{M}\}$, the initial state probability matrix $\pi =\{[\pi (i){]}_{N},\pi (i)=P(q_{i})\}$, the transition probability matrix $A=\{{\left[a_{ij}\right]}_{N\times N},a_{ij}=P(q_{j}|q_{i})\}$ and the emission probability matrix $B=\{{\left[b_{j}(k)\right]}_{N\times M},b_{j}(k)=P(o_{k}|q_{j})\}$ [24,25,26]. Where, N is the number of Q and M is the number of O. Q is the hidden state and O is the observable state. π represents the Q at t=1. A represents the transition probability between the states, indicating the probability that the state is true at t+1 when the state is true at t. B represents the probability that the O is true when the Q is true at t. We represented the HMM as λ=(A,B,π). Hypothesizing the hidden state is dependent on its previous moment only, and the observable state is dependent on the hidden state of the Markov chain [27]. An HMM can be represented as a probabilistic network (Figure 3).

Set of Variables

In this study, we took the driving intention at t as the hidden variable and running data of the vehicle at t as the observation variables. We assumed the driving behavior at t to be the driving intention at t−1 (t≥1). When $a_{c1}=a_{c1}^{1}$ at t, the driving intention of the following driver at t−1 was seen as deceleration; when $a_{c1}=a_{c1}^{2}$ at t, the driving intention at t−1 was seen as keeping speed; when $a_{c1}=a_{c1}^{3}$ at t, the driving intention at t−1 was seen as acceleration. The $<\Delta v_{e},\Delta d_{e},v_{1},v_{2},a_{2},d,\Delta v,\Delta a_{c}>$ formed the initial observed variables. We applied the factor analysis method to reduce the dimension of the original observation variables in SPSS 24 [28]. Table 3 shows the common factor variance of the initial observed variables. The sizes of extracted common factor variance for the parameters were very close, which indicated that the factor analysis method was practicable. Table 4 shows the interpretation of total variance in factor rotating. As we can see, the interpretation of total variance first three factors contained 93.86% of all information.

Figure 4 shows the information content of each factor. The gravel figure also shows that the information contained in the first three common factors meets the modeling requirements. The next step was to find three parameters that best represent the first three common factors.

Table 5 shows the rotation results of the factor load matrix. $\Delta d_{e}$ opened maximum weight projection (95.2%) on the main factor 1. d had the maximum weight (89.6%) in the factor space for factor 2. For the common factor 3, $\Delta v_{e}$ had the maximum weight (87.1%) in factor space. In summary, $\Delta d_{e}$, d and $\Delta v_{e}$ constituted the observations variables (factor set) of the driver’s intentions.

Model Training

The model training was to determine the value of A, B and π. The process of model training was, firstly, to initialize the model parameters λ={A,B,π} when the observation sequence $Y=(y_{1},y_{2},\cdots ,y_{T})$, state sequence $X=(x_{1},x_{2},\cdots ,x_{T})$, all possible sets of observations $\Phi =(\varphi_{1},\varphi_{2},\cdots ,\varphi_{M})$, and all possible sets of hidden states $S=(s_{1},s_{2},\cdots ,s_{N})$ were known. Secondly, we worked out the probability of the observation sequence P(Y|λ) through the forward–backward algorithm [29]. Finally, we determined the P(Y|λ) convergence and the model parameters through the Baum–Welch algorithm [30]. Figure 5 shows the model training process. We describe the key steps of model training in Appendix A. We used 68.5% of the original data to train the model.

### 2.2. Results

We obtained the probability of intentions by model training. The probabilities of deceleration (Int1), keeping speed (Int2) and acceleration (Int3) were 0.247, 0.468, and 0.285, respectively. Figure 6 shows the state transition probability (A). As can be seen from π and A, the probability of Int2 was significantly higher than that of acceleration and deceleration. This indicated that the results of model training were in line with the reality, because the driver would be driving in a stable state until the driving environment changed. The fact that the probabilities of Int1 and Int3 transferring to Int2 were larger than other transition probabilities also proved this. We show the probabilities of observation states (B) in Appendix A and discuss the change law of B in the discussion section.

### 2.3. Model Verification

We used the HMM to predict the driving intention. Scilicet, the driving intention at t (i.e., the driving behavior at t+1) could be predicted when the observation variables and the driving intention (acceleration) at t were known. We applied the rest of the data sets in the original database, which were unused in model training, to verify the model accuracy. The model accuracy in predicting the driving intention could be obtained by comparing the predicted value and the actual value of the vehicle acceleration at t+1. Figure 7 shows the model accuracy in predicting driving intentions. We used the approximate normal distribution method to calculate the confidence of the prediction accuracy [31], and the 95% confidence intervals of recognition accuracy for Int1, Int2 and Int3 were (0.761, 0.825), (0.845, 0.889) and (0.783, 0.849), respectively. The result showed that the accuracy of the driving intention prediction model proposed in this paper was acceptable compared with many existing driving intention prediction algorithms (the prediction accuracy is generally between 75% and 90%) [32,33,34].

## 3. Study 2-Driving Intention Prediction Models Adapting to Multi-Mode Emotions

### 3.1. Materials and Methods

#### 3.1.1. Experiment Design

The purposes of the experiments in this study were to collect the driving data of drivers in different emotional states. The experimental procedure related to driving was consistent with study 1. It was different in that the participants were in different emotional states.

Participants

The same experimental sample of 62 adult drivers volunteered to participate in this study.

Driving Emotion Classification and Selection

Psychologists classified emotions into different categories according to different criteria and theoretical constructions [35,36]. Among the many models of emotional structure, a revised edition of the Geneva Emotion Wheel was most suitable for the driving context [37]. This edition derived from the original wheel. Back-and-forth translation and the exchange of emotional qualities made the tool more suitable for traffic settings [38,39]. According to [40], pleasure was also a high-frequency driving emotion. We summarized the 18 emotions involved in [37,38,39,40] into a scale. The participants were asked to give a score from 0 to 5 for each emotion according to their experience during driving. Finally, 8 emotions with higher scores were selected, namely anger (Em_1_), surprise (Em_2_), fear (Em_3_), anxiety (Em_4_), helplessness (Em_5_), contempt (Em_6_), relief (Em_7_), and pleasure (Em_8_).

Experiment Materials

Aside from the driving experiment equipment mentioned in study 1, the experiment materials also referred to emotion induction materials in this study. Emotion induction materials included pictures and movie clips derived from the network, International Affective Picture System (IAPS) and Chinese Affective Picture System (CAPS). Figure 8 shows some emotional materials and Table 6 illustrates the movie clips applied in this study.

Emotion Induction

It is difficult to induce an individual’s emotions effectively and measure the level of emotional arousal accurately [40]. The methods most used in emotion induction research were movie clips [41], personalized recall [42], picture viewing [43], acoustic material [44] and standardized imagery [45]. Analysis results showed that music, movies and imagery were relatively ideal methods for emotion induction and the success rate was more than 75% [46,47]. Yet the driver, generally speaking, needs to spend most of his/her attention on the driving, which further increases the difficulty to induce the specific emotions needed. In order to effectively induce and maintain the driver’s emotions while driving, while minimizing the impact on driving tasks, we designed a new emotion induction process. The emotion induction process contained four steps (shown in Figure 9). The preliminary induction and in-depth induction carried out before the driving experiments, and the driving context assumption and emotional experience reminders implemented while the participants drove the car.

In the preliminary induction of anger, surprise, fear, and anxiety, we presented the pictures with specific emotional overtones and showed the contents expressed in the pictures to the participants, respectively. In the preliminary induction of helplessness and relief, we applied the personalized recall method to awake participants’ helplessness and relief emotion. For the helplessness, we guided the participants to recall their experiences of failure to pass the examination, unable to pay huge housing or car loan, etc. In the induction of relief, we guided the participants to describe their own experiences, such as passing the test of driving skill. For contempt induction, a scene of driving competition was set on the driving simulator in the virtual driving experiment. Firstly, we constructed a number of driving routes in the simulated driving scene, and obstacles, such as the construction area, were set up in the driving routes. Secondly, we ordered five experiment organizers and one participant to drive a virtual vehicle along the routes in sequence. Thirdly, the simulator controller controlled the driving competition to let the participant always reach the end of the experiment routes with the shortest time. Fourthly, we announced the driving competition results were to the participant and praised the driving skill of the participant to let him or her produce a strong sense of superiority. In actual driving experiments, we also used the personalized recall method to preliminarily induce contempt. The participants were guided to recall their experiences of feeling very superior, such as passing the course test with a higher score than their classmates, getting more appreciation or attention than others, etc. For the pleasure emotion, we informed the participants that they would get prizes if they finished the experiment. The in-depth inductions of emotions were carried out based on the preliminary inductions at the first opportunity. The induction materials for each emotion contained two or three movie clips. We picked one of the movie clips at random to play to the participants. In order to avoid the reduction in emotional induction effect caused by the advance understanding of the movie, the movie clips applied in the actual and virtual driving experiment for the same participant were not repetitive. The driving experiment started instantly after the above two steps. The driving task attributes had an important impact on the driver’s mood [48,49]. For example, in commuter driving, drivers were very likely to be anxious about the traffic congestion. Therefore, we assigned each driving experiment a hypothetical driving situation or task attribute to maintain the induced driving emotion. Table 7 shows the hypothetic driving task attributes for each emotion. During driving, we recalled the participants’ induced emotions through voice guidance, psychological suggestion or music (only for relief).

Driving Experiment

The equipment and procedures of the driving experiments were same as in study 1. Corresponding to each emotion state, each participant participated in eight actual driving experiments and eight virtual driving experiments. Unlike study 1, an important task while driving was to measure the driver’s emotional state. As an important branch of affective computing, emotion recognition patterns based on electroencephalogram (EEG) [50], electrocardiogram (ECG) [51], galvanic skin response (GSR) [52], respiration (RSP) [53], facial expression [54], and speech [55] have been the popular human emotion recognition methods. While we investigated eight kinds of driving emotions in this paper, and it seems impossible to find a measurement that effectively works for all of them with the mentioned methods. In view of these, we used the PAD emotion scale and model to measure the participants’ emotional states in the driving experiments. The PAD model was proposed by A. Mehrabian and J. Russe [56,57]. According to the model, emotions are composed of Pleasure–displeasure (P), Arousal–nonarousal (A), and Dominance–submissiveness (D). The values in each dimension range from −1 to +1 and the values can be used to represent specific emotions, which construct a workable framework for affective computing. Previous research results have shown that the three dimensions can effectively explained human emotions and the three dimensions are not limited to describing the subjective experience of emotions but also have a good mapping relationship with the external expression and physiological arousal of emotions [57]. Since the PAD model was proposed, many scholars have been working on building the mapping relationship between emotions and the three-dimensional space of PAD. The PAD space distributions of the 8 emotions in this paper are summarized in Figure 10 based on previous studies [58,59,60,61].

The PAD measurement tool, which is in line with the psychological measurement theory [28], is the PAD emotion scale (shown in Figure 11). As can be seen, each dimension is divided into nine states and five of these states are represented by cartoon figures with different shapes and sizes. There is a blank space between two cartoon figures in the same dimension. For the dimension “P”, the facial expressions of the cartoon figures are displayed one by one from extreme unhappiness on the far right (value-1), to the neutrality in the middle (value 0), and then to extreme happiness on the far left (value 1). For the dimension “A”, the size and irregularity of the image in the cartoon bodies and the size of the cartoon figures’ eyes represent different states from sleepy and indifferent on the far right (value-1), to relatively calm in the middle (value 0), and then to very excited on the far left (value 1). For the dimension “D”, the cartoon’ body size and eye size represent the degree of influence and control. It goes from being heavily influenced on the far right (value-1), to the normal state in the middle (value 0), and then to be able to control the situation well on the far left (value 1).

In order to be easy to use in the experiment and to facilitate the subsequent calculation, each dimension was noted from 0 to 100 based on the strength of each dimension and Table 8 shows the converted coordinates of each emotion. Before the emotion induction and driving experiments, we asked the participants to keep firmly in mind the meanings of PAD and make sure that they can accurately express their status in all dimensions without having to see the scale. The participants were required to report the PAD values (between 0 and 100 for P, A, and D, respectively) that matched their psychological state every minute. For the subjects, they can intuitively select a graphical representation of their current psychological state. The subjective and fuzzy description of one’s own emotional state, often applied in the emotional self-reports, was avoided. In the driving experiments, the participants also heard voice cues for emotion recall to maintain their induced emotions. We recorded the driving processes in real-time with the video detection system.

Each participant received eight kinds of induction. For a participant, the interval between the eight induction experiments must be long enough to ensure that his or her eight emotion experiences did not interfere with each other. We numbered the 62 participants and organized them to complete one kind of emotion induction and corresponding driving data acquisition in sequence according to the numbering order. Only after all participants completed the certain emotion induction and the corresponding driving data collection, the induction experiment of another emotion began. In the eight emotion-specific inductions, the order of 62 participants in the experiment was fixed. According to experiment records, we took the shortest time (11 days) to complete the emotion induction and corresponding driving data collection for all the participants in the relief virtual-driving experiment. This meant that the time interval between the two emotion induction experiments for the same participant was at least 11 days, and the eight emotional experiences did not interfere with each other. In addition, there were four steps involved in one emotional induction. In practice, we tried to shorten the time intervals between each step as much as possible to ensure that the emotional stimulation of each step had a positive impact on the emotional stimulation of the next step.

Effect Testing of Emotion Induction

In order to ensure the data were valid, we carried out the tests on the effectiveness of emotional induction. In the driving experiments, drivers reported PAD values for their emotional state every minute. Therefore, in each emotion state, we obtained 620 emotion indicator results in the actual and virtual driving experiment, respectively. Each ser of emotion indicator data represented the driver’s emotional state within a minute. We took the Euclidean distance between the reported coordinates (reported by the participants in the driving experiments) and converted coordinates of emotion in the PAD three-dimensional space as the measurable criteria of emotion activation [62]. The closer the distance was, the higher the emotion activation degree was. In the measurement tool of the PAD model, the coordinate difference value of each dimension state was 0.25. Therefore, Euclidean distance less than 0.25 from the reported PAD to the corresponding emotional coordinate point in the PAD original three-dimensional space indicated that the emotional state was effectively induced. According to this criterion, we obtained the percentage of effective emotional data on the total data for each emotion state (Figure 12) and the deleted non-effective emotional data.

#### 3.1.2. Model Construction

The driving intention prediction model, adapting to the eight kinds of emotions, were established. For each model, the process of data processing and model training was consistent with study 1. We used 70% of the corresponding experimental data for each emotional state for model training, and applied the remaining data to the corresponding model verification.

### 3.2. Results

We obtained the probability distributions of the three driving intentions in different emotional states and the transition probability between the three intentions in the proposed HMMs by model training. Figure 13 shows the training results. Compared to the natural state, there was no significant change in the probability distribution of Int2 under other emotion states, except for anxiety. In the state of anxiety, the probability of the driver keeping speed decreased significantly, while the probability of accelerating and decelerating both increased. In addition, the probability of acceleration increased dramatically. The same rule of probability increase and decrease also applied to the intention in the state of surprise emotion, but the change in probability was less obvious than that in the anxiety emotion. In the states of anger and contempt, the driver’s intention to decelerate reduced significantly, while the probability of accelerating intention increased observably, and the probability of keeping speed increased slightly. In fear and helplessness, the intention to accelerate significantly reduced, and the probability of maintaining speed increased to a certain extent. The probability of a driver to decelerate increased significantly in fear, but decreased significantly in helplessness. Aside from the fact that that the probability of Int1 increased significantly in the pleasure state, the probability distribution law of other driving intentions did not change significantly in the relaxed and pleasure state compared with the natural state. Corresponding to the change in the probability distribution, if the probability of one driving intention increased, the transition probability of the other two driving intentions to it would increase too. On the contrary, if the probability of one driving intention reduced, the transition probability of the other two intentions to it would reduce. We showed the probabilities of observation states for different emotions in Appendix A and discussed the change in the law of observation sequence in the discussion sections.

### 3.3. Model Verification

We used the HMMs proposed in this study to predict the driving intention adapting to driver’s different emotional states. We applied the rest of the data sets, which were unused in model training, to verify the accuracy of the proposed models. We obtained the accuracy of the models in predicting the driving intention by comparing the predicted value and the actual value of the vehicle acceleration. Figure 14 shows the accuracy in predicting driving intentions of the HMM for different emotions. The results showed that the prediction accuracy for the deceleration, keeping speed, and acceleration intentions in different models reached a similarly acceptable level.

## 4. Discussion

In studies 1 and 2, we obtained and analyzed the probability distribution and transition probability of the three driving intentions in drivers’ natural states and different emotional states.

In order to examine whether emotions had significant influences on the probability distribution of the three intentions, we used the *t*-test method [31] to compare the probability of driving intention in the non-emotional state (Ne) with that in the eight emotional states. In this process, we merged the actual and virtual driving data of each participant to be one data sample. Table 9 shows the *t*-test results. As can be seen from Table 9, the influences of anxiety and relief on the probability of Int1 were not significant. The effect of other emotions on the probability of Int1 were significant. The influences of anger, surprise, fear and relief on the probability of Int2 were not significant. The effect of other emotions on the probability of Int2 were significant. Except for pleasure and relief, the effect of emotions on the probability of Int3 were significant.

Analogously, we used the *t*-test method to examine the significance of emotional influence on the transition probability of the three intentions and the test result are shown in Table 10. It can be seen that, except for the fact that relief had no significant effect on the transition of Int3, different emotions had significant impacts on the transition of driving intention (in whole or in part).

Furthermore, we applied the F-test [31] to verify whether there is a difference in the impact of the eight emotions on the transition probability of driving intentions. The test results are shown in Appendix A, and the results supported the conclusion that there were significant differences in the influence of eight emotions on the driving intention transition probability. We also used the q-test [31] to compare the effects of any two emotions on the transition probability of intention. The test results are shown in Appendix A.

Finally, we expected to obtain the changes in driving intention generated by the change in emotional the variable under the condition that other variables were unchanged or nearly unchanged. In the intention identification model constructed above, we got the probability distribution of driving environment parameters under different intention states ($B=\{{\left[b_{j}(k)\right]}_{N\times M},b_{j}(k)=P(o_{k}|q_{j})\}$). We used the Bayesian formula (shown in (1)) to solve the probability distribution of driving intention under different driving environment parameters.
(1)$$P\left(q_{i}|O\right)\frac{P(q_{i})P(O|q_{i})}{\sum_{j=1}^{N}P(q_{j})P(O|q_{j})}$$

In this study, the observation variable contained three parameters: $\Delta d_{e}$, d and $\Delta v_{e}$. $\Delta d_{e}$ represented the difference between the actual following distance and the driver’s expected distance and $\Delta v_{e}$ represented the difference between the actual vehicle speed and the driver’s expected speed. In the process of driving, drivers often express their psychological needs through speed and distance—i.e., the pursuit of higher speed and shorter distance [23,63]. $\Delta v_{e}^{1}$ represented that the driver’s speed need was not being met; $\Delta v_{e}^{2}$ indicated that the current vehicle speed was just enough to meet the driver’s speed need; $\Delta v_{e}^{3}$ represented that the speed of the vehicle was too great for the driver’s speed need. Similarly, $\Delta d_{e}^{3}$ represented that the driver’s distance need was not being met; $\Delta d_{e}^{2}$ indicated that the current following distance was just enough to meet the driver’s distance need; $\Delta d_{e}^{1}$ represented that the following distance was too short for the driver’s distance need. Considering the basic definitions of $v_{e}$ and $d_{e}$ [64,65], the combination variable of $\Delta d_{e}$ and $\Delta v_{e}$ was defined as the satisfaction degree (Sd). The component of Sd in the speed dimension ($Sd_{1}$) was assigned to 0, 1, and 2, respectively, when the $\Delta v_{e}$ was $\Delta v_{e}^{1}$, $\Delta v_{e}^{2}$, and $\Delta v_{e}^{3}$, respectively. The component of Sd in the distance dimension ($Sd_{2}$) was assigned to 0, 1, and 2, respectively, when the $\Delta d_{e}$ was $\Delta d_{e}^{3}$, $\Delta d_{e}^{2}$, and $\Delta d_{e}^{1}$, respectively. When the sum of $Sd_{1}$ and $Sd_{2}$ was 0 or 1, the value of Sd was assigned to 1, indicating that the driver’s demand satisfaction was low. When the sum of $Sd_{1}$ and $Sd_{2}$ was 2, the value of Sd was assigned to 2, indicating that the driver’s demand satisfaction was good. When the sum of $Sd_{1}$ and $Sd_{2}$ was 3 or 4, the value of Sd was assigned to 3, indicating that the driver’s demand satisfaction was over met. In the car-following theory, the distance between vehicles is often regarded as an index to measure the safety risk [66]. In this study, the d was defined as risk degree (Rd) and the Rd was assigned to 3, 2, and 1, respectively, when the $\Delta_{d}$ was $d^{1}$, $d^{2}$, and $d^{3}$, respectively. The greater the value of Rd, the greater the risk of driving safety in the distance dimension. In order to show the change in driving intention probability, caused by driving emotion evolution more intuitively, (2) calculates the change rates of driving intention probability under different emotions.
(2)$$Gr=\frac{P(Int_{i}|W_{n},Em_{j})-P(Int_{i}|W_{n})}{P(Int_{i}|W_{n})}\times 100\%\;i=1,2,3\;j=1,2,\cdots ,8\;n=1,2,\cdots ,9$$
where, Gr is the change rate of driving intention probability. $Int_{i}$ is the driving intention. $W_{n}$ denotes the combinatorial states of $Em_{j}$, Sd and Rd. $Em_{j}$ is the emotional state.

Through the above processing and calculation, we got the change rates of the driving intention probability between the driver’s anger and natural state. As can be seen Figure 15a, compared to the natural state, the probability of acceleration significantly increased while the probability of deceleration significantly decreased in their anger emotional state. Additionally, the degree of the above changes decreased with the increase in Sd or Rd. Many previous researchers have proven that the drivers in the anger emotion prefer aggressive driving behavior [10,67,68]. The results in this study showed that drivers were more inclined to accelerate when they were in an angry state—i.e., preferred to seek a higher driving speed and a shorter following distance. Figure 15b shows the change rates of the driving intention probability between the driver’s surprise emotion and the natural state. It was shown that no remarkable change in driving intention probability took place when the Sd was at a high level. In the moderate or low Sd, the probability of keeping speed decreased significantly and the probability of the other two intentions increased observably when the Rd was moderate or low. Surprise promoted the cognitive control of events and the ability to adapt to sudden environmental changes [69,70]. Therefore, surprise lead to an increase in the sensitivity of drivers to external environmental changes and made drivers easily attracted to other environmental factors. The increased cognitive load could make the driver’s vehicle control strategy confused, thus making the driver change the speed frequently within a certain range.

Figure 16a shows the change rates of the driving intention probability between the fear and natural state. It is evident that the probability of the intention to decelerate increased while the probability of the driver’s intention to accelerate decreased. When Sd was moderate, the change was obvious only when the Rd was high. The changes were more conspicuous when the Sd was high and low, and the change rates increased with the increase in Rd. We see the opposite rule of change rates in fear compared with anger. The driver was more likely to decelerate—i.e., drivers preferred to choose conservative driving behaviors. We can find similar results in many previous studies [71,72]. It is generally accepted that fear increase the driver’s risk perception [73], and the drivers prefer to choose the driving pattern with low risk [74]. Figure 16b shows the change rates of the driving intention probability between anxiety and natural state. As can be seen from Figure 16b, the probability of the intention to accelerate and decelerate both markedly increased, while the probability of keeping speed significantly decreased. Many previous studies have shown that anxiety increases driving risks [75,76,77], and our findings support that, considering constant acceleration or deceleration could increase the risk of driving safely beyond dispute.

Figure 17a shows the change rates of the intention probability between the driver’s helplessness and the natural state. A probabilistic variation, similar to that of fear, appeared in helplessness, while the intention probability distribution changed to a lower degree under the helplessness emotion. Helplessness was proven to be an emotion similar to fear [73]. Figure 17b shows the change rates of the driving intention probability between the driver’s contempt emotional state and the natural state. Few previous studies have focused on driver contempt in the field of driving behavior, while the results of this paper have shown that the probabilistic change in driving intentions was similar to anger but less variable. The probability of the driver’s intention to accelerate significantly increased, while the probability of the driver’s intention to decelerate significantly decreased. Additionally, the degree of the above changes decreased with the increase in Sd or Rd.

Figure 18a shows the change rates of the driving intention probability between the driver’s relief emotional state and the natural state. As can be seen, the change of driving intention probability caused by relief was insignificant. This seemed to suggest that a driver made rational driving decisions in a relaxed emotional state. However, some researchers pointed out that relief increased driving risk by reducing driver’s awareness of risk [78]. Figure 18b shows the change rates of the driving intention probability between the driver’s pleasure emotion state and the natural state. The probabilistic variation of driving intention in the pleasant emotional state was similar to that of relief, while the change degree in the pleasant emotional state increased slightly.

## 5. Conclusions

This study focused on the influence of driver’s different emotional states on the generation and transition of driving intention states. Through constructing the driving intention prediction models suitable for the driver’s natural state, and eight typical emotional states, respectively, we obtained the probability distribution and transition probability of three driving intentions in a driver’s different states. By controlling a single variable, we obtained the generation probability of driving intention under the condition that the emotional state changed while the driving environment parameters remained. Finally, we compared the driving intention generation probabilities in different emotional states with those in natural states and got the differences in driving intention transitions caused by the driver’s emotion evolutions. The basic idea of most previous driving behavior prediction models is generally to predict the driver’s behavior based on the current running state of the vehicle and the time–space relationship with other vehicles in the vehicle aggregation situation (relative speed, relative distance, etc.). The premise of this behavior prediction is that drivers can always effectively perceive the driving environment information and make rational analysis of the traffic safety situation based on the information in the driving behavior decision. In this case, the driving behavior prediction model is likely to make wrong decisions because it ignores the driver’s current mental state. For example, when a driver is in the anger emotion, he or she is very likely to have unreasonable perceptions of the interaction between vehicles (taking other drivers’ normal overtaking behavior as a provocation to themselves, etc.), and then adopt aggressive driving behavior. Obviously, if the behavior prediction model still follows the conventional model to predict the behavior, it will inevitably get results that are not consistent with the actual situation. In contrast, if the driver’s current emotional state of anger can be fully considered in the driving behavior prediction, and then realize that the probability of the driver taking aggressive driving behavior will be greatly increased, the accuracy of behavior prediction will be greatly improved. When it needs to be pointed out, the premise of constructing an emotional driving behavior prediction model is that the on-board intelligence system can accurately recognize the emotional state of the car driver. Finally, yet importantly, it is necessary to point out some limitations of this study. The first is the hypothesis of driving intention and emotion. In the process of constructing, the intention prediction model, based on the hidden Markov model, we assumed the transfer of driving intention as a Markov chain. In the induction and measurement of participants’ emotions, we assumed that the PAD accurately depicted the human emotions, and we deemed the eight emotions in this study to have mapping relationships with the three dimensions of PAD. In addition, as an exploratory study, the car-following driving environment with a single scene was selected, which resulted in the research results of this paper could not being applicable to a more complex driving environment.

## Figures and Tables

**Figure 1 ijerph-17-06962-f001:**
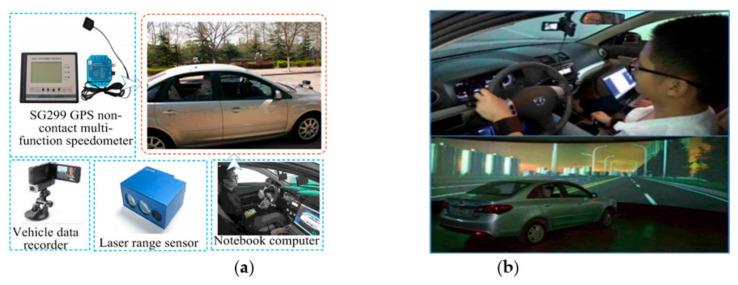
Experiment equipment. (**a**) Actual experiment vehicle; (**b**) Virtual driving platform.

**Figure 2 ijerph-17-06962-f002:**
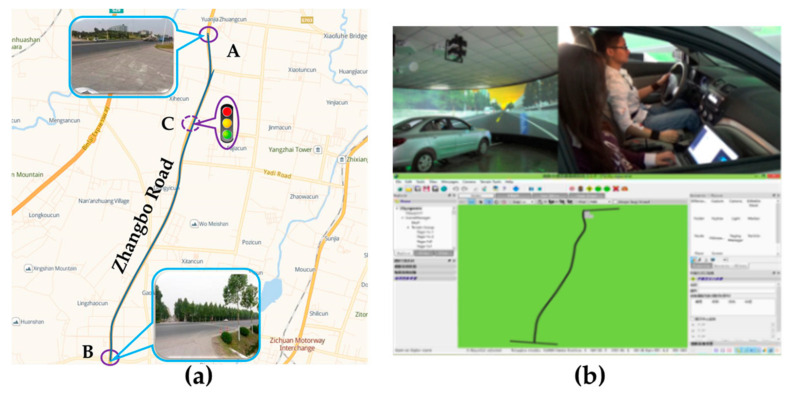
Experiment route. (**a**) Actual driving route; (**b**) Virtual road section.

**Figure 3 ijerph-17-06962-f003:**
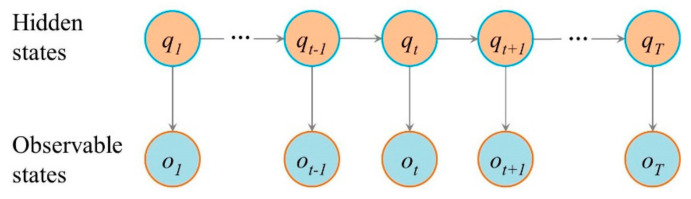
An HMM represented as a probabilistic network.

**Figure 4 ijerph-17-06962-f004:**
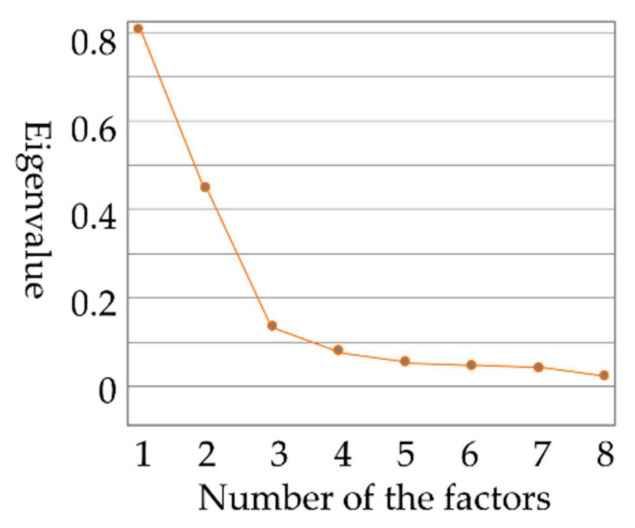
Scree plot of the information content of each factor.

**Figure 5 ijerph-17-06962-f005:**
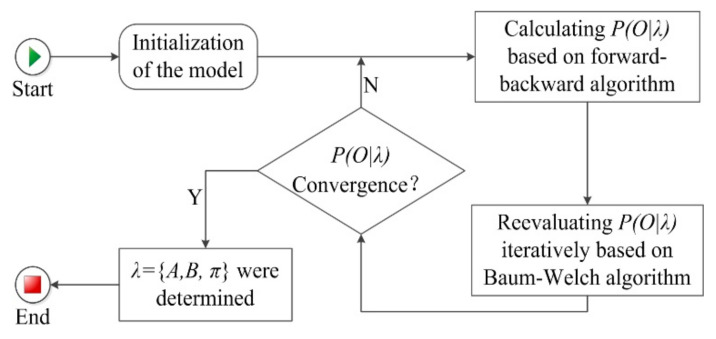
Model training process.

**Figure 6 ijerph-17-06962-f006:**
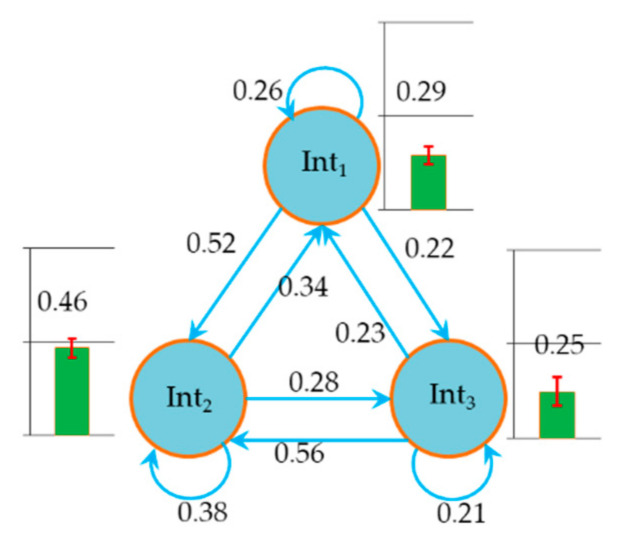
Transition probability between driver’s intention states in the HMM.

**Figure 7 ijerph-17-06962-f007:**
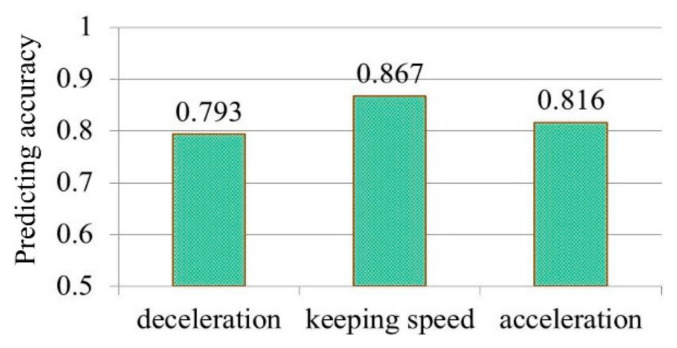
Accuracy in predicting driving intentions of proposed HMM.

**Figure 8 ijerph-17-06962-f008:**
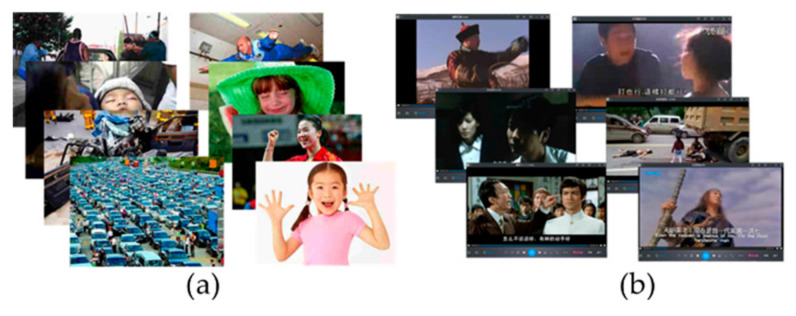
Some emotion induction materials. (**a**) Pictures; (**b**) Movie clips.

**Figure 9 ijerph-17-06962-f009:**
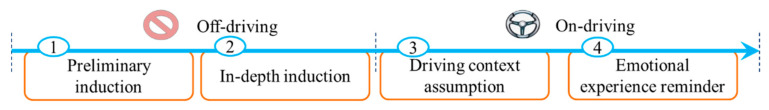
Emotion induction process.

**Figure 10 ijerph-17-06962-f010:**
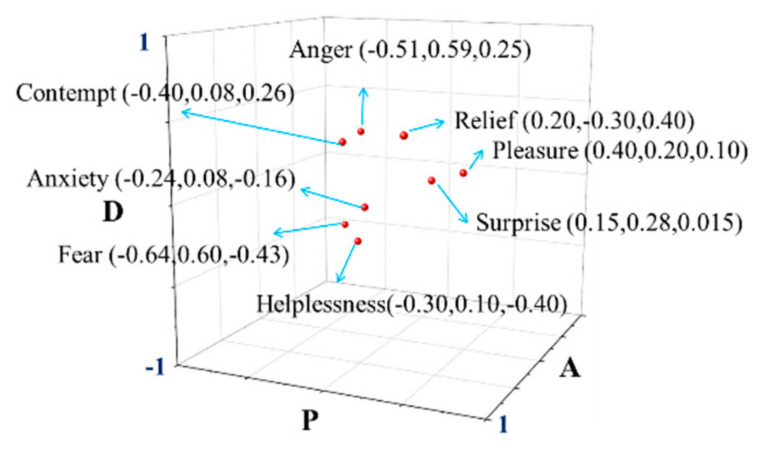
Distribution of the eight emotions in PAD space.

**Figure 11 ijerph-17-06962-f011:**
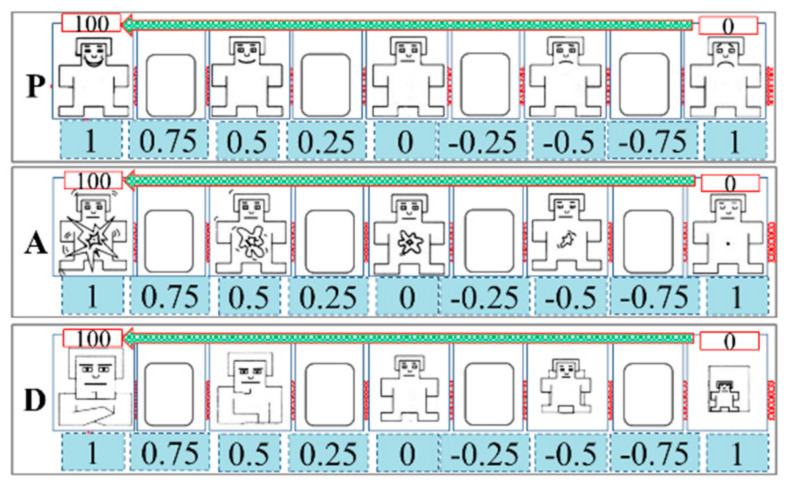
PAD emotion scale.

**Figure 12 ijerph-17-06962-f012:**
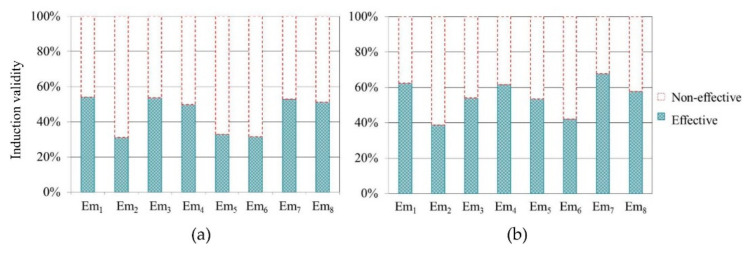
Emotion measurement results. (**a**) Actual driving experiments; (**b**) Virtual driving experiments.

**Figure 13 ijerph-17-06962-f013:**
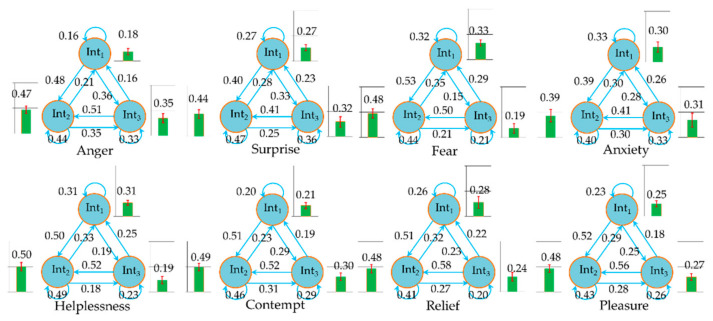
Transition probability between driver’s intention states in the different HMMs.

**Figure 14 ijerph-17-06962-f014:**
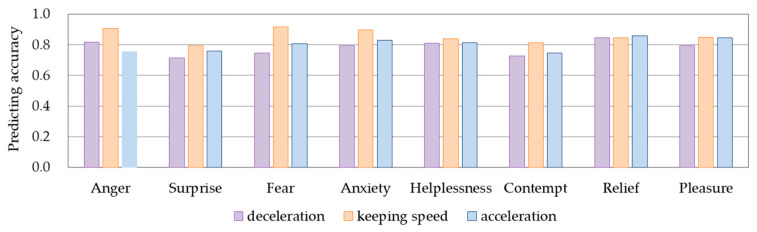
Accuracy in predicting driving intentions of HMMs for different emotions.

**Figure 15 ijerph-17-06962-f015:**
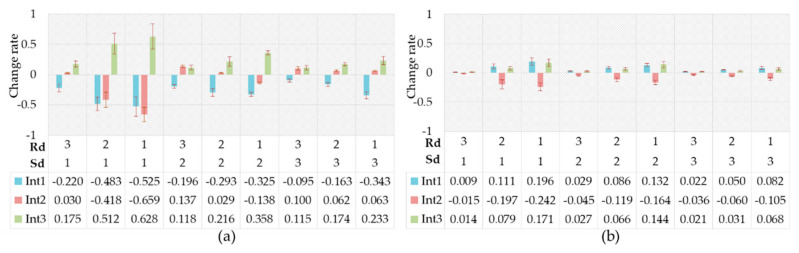
Change rates of driving intention probability. (**a**) Caused by anger; (**b**) Caused by surprise.

**Figure 16 ijerph-17-06962-f016:**
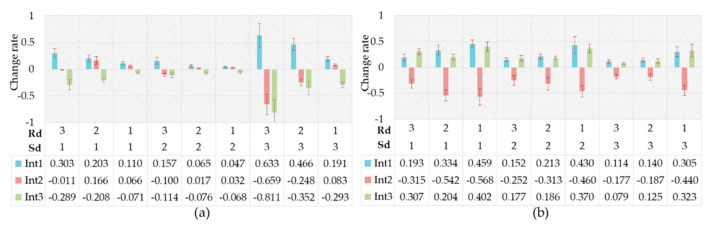
Change rates of driving intention probability. (**a**) Caused by fear; (**b**) Caused by anxiety.

**Figure 17 ijerph-17-06962-f017:**
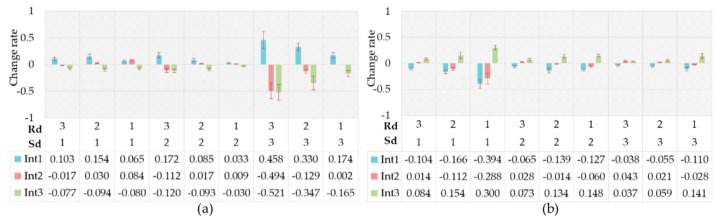
Change rates of driving intention probability. (**a**) Caused by helplessness; (**b**) Caused by contempt.

**Figure 18 ijerph-17-06962-f018:**
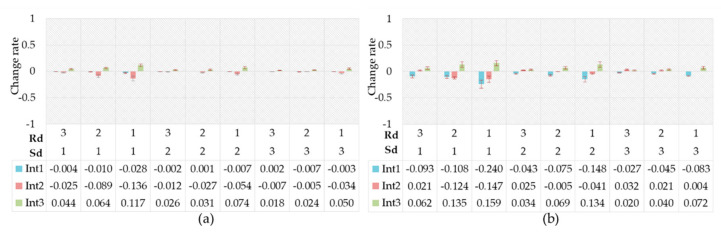
Change rates of driving intention probability. (**a**) Caused by relief; (**b**) Caused by pleasure.

**Table 1 ijerph-17-06962-t001:** Variables and symbols.

Symbol	Variable	Symbol	Variable
$v_{e}$	expected driving speed (km/h)	$a_{c2}$	acceleration of leading car (m/s^2^)
$d_{e}$	expected following distance (m)	d	relative distance between (m)
$v_{1}$	velocity of following car (km/h)	Δv	relative velocity (km/h)
$a_{c1}$	acceleration of following car (m/s^2^)	$\Delta a_{c}$	relative acceleration (m)
$v_{2}$	velocity of leading car (km/h)		

**Table 2 ijerph-17-06962-t002:** Rules for data discretization.

Item	Symbol	Attribute	Value	Item	Symbol	Attribute	Value	Item	Symbol	Attribute	Value
$\Delta v_{e}$	$\Delta v_{e}^{1}$	$\Delta v_{e}<-5$	1	$v_{2}$	$v_{2}^{1}$	$v_{2}<55$	1	$a_{c1}$	$a_{c1}^{1}$	$a_{c1}<-0.5$	1
	$\Delta v_{e}^{2}$	$-5<\Delta v_{e}<5$	2		$v_{2}^{2}$	$55<v_{2}<60$	2		$a_{c1}^{2}$	$-0.5<a_{c1}<0.5$	2
	$\Delta v_{e}^{3}$	$\Delta v_{e}>5$	3		$v_{2}^{3}$	$v_{2}>60$	3		$a_{c1}^{3}$	$a_{c1}>0.5$	3
$\Delta d_{e}$	$\Delta d_{e}^{1}$	$\Delta d_{e}<-5$	1	d	$d^{1}$	*d* < 8	1	$a_{c2}$	$a_{c2}^{1}$	$a_{c2}<-0.5$	1
	$\Delta d_{e}^{2}$	$-5<\Delta d_{e}<5$	2		$d^{2}$	8<d<13	2		$a_{c2}^{2}$	$-0.5<a_{c2}<0.5$	2
	$\Delta d_{e}^{3}$	$\Delta d_{e}>5$	3		$d^{3}$	*d* > 13	3		$a_{c2}^{3}$	$a_{c2}>0.5$	3
$v_{1}$	$v_{1}^{1}$	$v_{1}<55$	1	Δv	$\Delta v^{1}$	Δv<−5	1	$\Delta a_{c}$	$\Delta a_{c}^{1}$	$\Delta a_{c}<-1.5$	1
	$v_{1}^{2}$	$55<v_{1}<60$	2		$\Delta v^{2}$	−5<Δv<5	2		$\Delta a_{c}^{2}$	$-0.15<\Delta a_{c}<0.15$	2
	$v_{1}^{3}$	$v_{1}>60$	3		$\Delta v^{3}$	Δv>5	3		$\Delta a_{c}^{3}$	$\Delta a_{c}>0.15$	3

**Table 3 ijerph-17-06962-t003:** Common factor variance.

Factor	Initial Value	Extracted Value	Factor	Initial Value	Extracted Value
$\Delta v_{e}$	1.000	0.983	$a_{c2}$	1.000	0.732
$\Delta d_{e}$	1.000	0.921	*d*	1.000	0.947
$v_{1}$	1.000	0.975	Δv	1.000	0.933
$v_{2}$	1.000	1.000	$\Delta a_{c}$	1.000	0.834

**Table 4 ijerph-17-06962-t004:** Interpretation of total variance.

	Initial Eigenvalue			Extracted Loads of Sum Squares			Rotated Loads of Sum Squares		
	T ^1^	V% ^2^	CV% ^3^	T	V%	CV%	T	V%	CV%
1	8.02	35.30	35.30	8.93	35.30	35.30	7.49	34.15	34.15
2	4.74	31.79	67.09	4.65	31.79	67.09	3.81	29.67	62.32
3	1.34	26.77	93.86	1.74	26.77	93.86	1.14	30.04	93.86
4	0.91	2.40	96.26						
5	0.67	1.35	97.61						
6	0.53	1.03	98.64						
7	0.45	0.84	99.48						
8	0.32	0.52	100						

^1^ T represents the total eigenvalue. ^2^ V represents the variance. ^3^ CV represents the cumulative variance.

**Table 5 ijerph-17-06962-t005:** Rotated factor matrix.

Factor	Component			Factor	Component		
	1	2	3		1	2	3
$\Delta v_{e}$	0.061	0.100	0.871	$a_{c2}$	0.004	0.768	0.103
$\Delta d_{e}$	0.952	0.039	0.203	d	0.880	0.896	0.064
$v_{1}$	−0.004	−0.633	0.370	Δv	0.008	0.113	0.805
$v_{2}$	0.710	0.048	0.057	$\Delta a_{c}$ *_c_*	0.010	0.837	0.061

**Table 6 ijerph-17-06962-t006:** Movie clips for emotion induction.

Emotion	Movie Clips
Anger	1. Title: Fist of Fury/Source: CAPS/Duration: 4′18″ 2. Title: Video highlights on baby abuse/Source: Internet/Duration: 5′54″
Surprise	1. Title: Fabulous experiment/Source: Internet/Duration: 2′03″ 2. Title: Shocking news/Source: Internet/Duration: 3′32″ 3. Title: Guinness world records highlights/Source: Internet/Duration: 6′11″
Fear	1. Title: I hear too/Source: CAPS/Duration: 10′35″ 2. Title: Vicious/Source: Internet/Duration: 11′36″
Anxiety	1. Title: Curve/Source: Internet/Duration: 9′51″ 2. Title: Anxious Chinese/Source: Internet/Duration: 3′31″
Helplessness	1. Title: Fauve/Source: Internet/Duration: 16′24″ 2. Title: The Mist/Source: Internet/Duration: 4′09″
Contempt	1. Title: Hobble seats on high-speed train/Source: Internet/Duration: 4′44″ 2. Title: An old man who depends on his elders/Source: Internet/Duration: 3′21″
Relief	1. Title: World beauty appreciation/Source: Internet/Duration: 6′02″ 2. Title: Dancing Fluorescent Grass/Source: Internet/Duration: 6′10″
Pleasure	1. Title: Eat Hot Tofu Slowly/Source: CAPS/Duration: 1′29″ 2. Title: The Eagle Shooting Heroes/Source: CAPS/Duration: 1′02″ 3. Title: Mr. Bean/Source: Internet/Duration: 6′45″

**Table 7 ijerph-17-06962-t007:** Hypothetic driving task attributes.

Emotion	Driving Task Attributes
Anger	1. Driving safety is maliciously violated. 2. Seeing other drivers improper driving deliberately. 3. In faced with bad traffic.
Surprise	1. Driving in an unfamiliar environment. 2. In case of emergency (no security threat is involved).
Fear	1. Almost caused an accident. 2. Witness the occurrence or the scene of a traffic accident. 3. Driving in extreme weather conditions.
Anxiety	1. Suffer from traffic jam on the way to work. 2. Lacking in self-confidence in faced with complex environment. 3. Driving with physically tired or mental stress.
Helplessness	1. About to face punishment for violating regulations. 2. Stuck in bad traffic.
Contempt	1. Seeing other drivers’ unskilled driving. 2. Seeing other drivers get punished for breaking regulations. 3. Driving performance is significantly better than others are.
Relief	1. Driving in good road and traffic conditions. 2. Getting rid of bad driving environment and into a better environment.
Pleasure	1. Driving in excellent road and traffic conditions. 2. Driving home after a long day at work. 3. Driving to the vacation destination. 4. Nice interaction with entertainment system during driving.

**Table 8 ijerph-17-06962-t008:** Converted coordinates of each emotion.

Emotion	P	A	D	Emotion	P	A	D	Emotion	P	A	D	Emotion	P	A	D
Anger	24.5	79.5	62.5	Fear	18	80	28.5	Helplessness	35	55	30	Relief	60	35	70
Surprise	57.5	64	50.75	Anxiety	38	54	42	Contempt	30	54	63	Pleasure	70	60	55

**Table 9 ijerph-17-06962-t009:** *t*-test for probability distribution of driving intention.

Paired Comparison	Int1		Int2		Int3	
	*t*^1^ Value	Sig. ^2^	*t* Value	Sig.	*t* Value	Sig.
Ne-Em1	11.09	***	−1.06	-	−8.58	***
Ne-Em2	2.12	**	1.60	-	−4.31	***
Ne-Em3	−4.04	***	−1.55	-	3.89	***
Ne-Em4	−0.70	-	4.27	***	−3.09	***
Ne-Em5	−2.32	**	−3.23	***	4.36	***
Ne-Em6	8.12	***	−2.58	**	−3.62	***
Ne-Em7	0.63	-	−1.66	-	0.66	-
Ne-Em8	4.18	***	−1.76	*	−1.74	*

^1^*t*_α = 0.1,61_ = 1.671, *t*_α = 0.05,61_ = 2, *t*_α = 0.01,61_ = 2.66. ^2^ *** represents the significance level 0.01, ** represents the significance level 0.05, * represents the significance level 0.1, - represents non-significance.

**Table 10 ijerph-17-06962-t010:** *t*-test for transition probability of driving intention.

**Paired Comparison**	**Int1 → Int1**		**Int1 → Int2**		**Int1 → Int3**	
	***t*^1^ Value**	**Sig. ^2^**	***t* Value**	**Sig.**	***t* Value**	**Sig.**
Ne-Em1	14.41	***	3.46	***	−8.56	***
Ne-Em2	−1.11	-	7.94	***	−6.11	***
Ne-Em3	−3.85	***	2.78	***	−0.74	-
Ne-Em4	−5.22	***	11.85	***	−7.93	***
Ne-Em5	−2.80	***	4.98	***	−2.99	***
Ne-Em6	8.22	***	2.82	***	−6.48	***
Ne-Em7	0.40	-	1.98	*	−1.94	*
Ne-Em8	3.87	***	0.49	-	−2.00	**
**Paired Comparison**	**Int2 → Int1**		**Int2 → Int2**		**Int2 → Int3**	
	***t* Value**	**Sig.**	***t* Value**	**Sig.**	***t* Value**	**Sig.**
Ne-Em1	14.44	***	−5.29	***	−4.54	***
Ne-Em2	5.88	***	−7.85	***	2.72	***
Ne-Em3	2.16	**	−4.15	***	1.65	-
Ne-Em4	6.73	***	0.52	-	−5.42	***
Ne-Em5	4.06	***	−7.35	***	3.63	***
Ne-Em6	11.95	***	−4.21	***	−5.32	***
Ne-Em7	2.05	**	−2.54	**	0.44	-
Ne-Em8	5.44	***	−3.02	***	−1.49	-
**Paired Comparison**	**Int3 → Int1**		**Int3 → Int2**		**Int3 → Int3**	
	***t* Value**	**Sig.**	***t* Value**	**Sig.**	***t* Value**	**Sig.**
Ne-Em1	10.45	***	15.15	***	−17.94	***
Ne-Em2	0.06	-	8.77	***	−7.70	***
Ne-Em3	−5.81	***	5.96	***	−3.02	***
Ne-Em4	−1.74	*	13.87	***	−11.65	***
Ne-Em5	−1.33	-	5.06	***	−4.06	***
Ne-Em6	1.26	-	5.18	***	−5.21	***
Ne-Em7	0.30	-	−1.42	-	1.22	-
Ne-Em8	6.18	***	−0.07	-	−1.93	*

^1^*t*_α = 0.1,61_ = 1.671, *t*_α = 0.05,61_ = 2, *t*_α = 0.01,61_ = 2.66. ^2^ *** represents the significance level 0.01, ** represents the significance level 0.05, * represents the significance level 0.1, - represents non-significance.

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
