# Peer review of "Differences in Driving Intention Transitions Caused by Driver’s Emotion Evolutions"

_ijerph, 2020, doi:10.3390/ijerph17196962_

Round 1

Reviewer 1 Report

The paper presents methods to identify driving intention transition under the influence of emotions. In Study 1, a hidden markov model (HMM)-based method was developed for intention transition identification under normal driving conditions. In Study 2, the method was further applied to a series of typical driving emotions. The study of human factors and driving intentions is always difficult, but the authors did a remarkable job by carefully design the experiments and analyzing the data to derive novel observations. The paper is well-structured and easy to follow. Related work is properly referred and cited in the paper. 

Although the paper was well-written, there are still minor wording and grammatical problems here and there. To name a few: (i) Lack of spaces on Line 224; (ii) Line 230, "were into a scale"; (iii) Line 342 "t" and "he"; (iv) Line 456 and Line 457, no "the" in front of "Sdï¼›(v) Line 500 - Line 502, avoid using such a long sentence, it it really confusing. I therefore suggest the authors to have a professional English editor to further polish the paper. 

Annotations of the y-axis are needed for Figure 7, Figure 12, Figure 14 - Figure 18 are needed. 

Reviewer 2 Report

In their study, Liu and Wang investigate the impact of emotion on driving behavior. More specifically, they are interested in the question how emotional states change driving intentions. Using both a real-world and virtual driving setting, they report that different emotional states influence the driving behavior.

While I think that this study focusses on a relevant topic for both basic as well as applied research, I see a few major problems with the theoretical foundation, the design of the experiment as well as the analysis of the obtained data.

MAJOR CONCERNS:

Regarding the design of the study, the authors claim to induce emotion via a “new” emotion induction process (p.9 l.251). The authors seem to be aware of the difficulty in inducing emotions in human participants and therefore chose a two-step procedure that incorporates both an off-driving and on-driving component. While this procedure is appropriate for the goal of emotion induction, the authors investigate eight different emotions. Besides the fact that eight emotions are quite a lot, they use a within-subjects design, so that all participants experienced all emotions. Crucially, there is no information on experimental balancing/randomization procedures and/or induction protocols (e.g., time between inductions). This makes it difficult to accurately interpret the reported results of this study. Additionally, the emotion induction procedure is not standardized. In the off-driving phase, the eight emotion are induced using five different methods. In the on-driving phase, they used three additional methods for the emotion recall. Given the application of multiple induction methods between emotion and the proximity/similarity of emotions (e.g., fear and anxiety), it is impossible to validly compare conditions, especially considering the low degree of “effective” emotion measurement results around 50% (Figure 12 on p.13).

Regarding the analysis of the study, the authors use the recorded data from a (variant of the) widely used car-following (i.e. platooning?) paradigm to obtain speed, acceleration and distance measures. These measures are then used to inform a state-of-the-art Hidden Markov Model (HMM). The HMM takes the observed data and derives the (hidden) driving intentions. Although the authors report significant differences in driving intentions between conditions (e.g., p.7 l.196; p.13 l. 396ff), they do not report any statistical test results. In line with this, the interpretation of percentages for model verification as "reach[ing] a high level" (p.7 l.213, see also p.14 l.394) seems speculative without a statistical analysis or benchmark.

As a further major concern of the paper, the authors do not provide a clear link between the results of their study and their claim that “the results of this study could greatly improve the accuracy of driving behavior prediction” (p.17 l. 505f).

MINOR CONCERNS:

p.1 l.44: Does the term “labor load” refer to workload, a concept commonly used in psychological research? How is this linked to the concept of “cognitive load” (p.15 l.449)? Please use clear wording and provide links between related concepts.

p.4 l. 132f: On which basis was the “time window for a single state” set to 10 seconds? Without proper citation, this seems to be an arbitrary duration, that can massively alter results.

p.4 l.141: The variable “a” in the formula is ill/double-specified, as the same letter is also used as a measure of acceleration (e.g., Table 1 on p.4).

p.7 l.208: The information and justification for the division in training and test set is missing. However, it is provided for the second experiment (70% p.13 l. 361). On which basis was this division introduced?

p.12 l.338f: How was the confirmation of emotion reports after re-watching used within the study? Is this a common method? If so, please provide citation.

  1. 14 l.382ff: Please provide further clarification on the formal conflict between “increase” and “reduce”.

p.14 l.397ff: To me, the discussion seems to be an extended results section. Furthermore, please provide further information on the derivation of the concepts of demand satisfaction (Sd) and risk degree (Rd). These seem to be highly complex concepts, that are operationalized (supposedly) overly simplistic and without any theoretical justification in the study.

All over the manuscript the terms “could” and “would” are used incorrectly and/or out of context.

Reviewer 3 Report

The paper is really interesting. The relationship between driving emotion and behaviour worth to be investigated in order to improve driving behaviour prediction technology and active safety early warning system. The result of this study can represent an interesting starting point in the improvement of driving behaviour prediction.

found the paper to be overall well written and much of it to be well described. I believe the authors performed careful and thorough field and numerical processing.

I really appreciate reading this paper but it has some room for improvement. The main suggestions for improving the paper are reported below.

MAJOR COMMENTS:

  1. Introduction:
  • The literature review includes relevant papers. However, I found the introduction not straight forward, and instead of get directly to the point, which is interesting, as I wrote before, is vague and does not really underline why you are writing this paper, what are the needs and the target.
  1. Study 1-A Driving Intention Prediction Model Based on HMM:
  • From the description of the driving experiment, it is not clear if other vehicles were present during the driving experiment (apart from the vehicles used from the driving experiment). The presence of other vehicles could be a significant confounding with respect to the measurements of the behavioural variables. Could the authors elaborate on that?
  • Line 116: With reference to the sentence “due to regional conditions, the probability of signalized intersection has been minimized” could the authors explain this better?
  1. Study 2- Driving Intention Prediction Models Adapting to Multi-Mode Emotions:
  • Lines 281-286: Could the authors elaborate more on the meaning of “hypothetical driving situation or task attribute“? Could the authors explain better the steps 3 and 4 of the emotion induction process shown in Figure 9?
  1. Discussion:
  • Lines 400-401: With reference to the sentence “And we obtained the general influence law of driver’s different emotional states on driving intention could be obtained.” could the authors please reformulate or explain it better?
  1. Conclusion:
  • I found the conclusion overall well written. However, in my view, it would be appropriate to have a more detailed discussion on the practical implications of the study, which I think could be very interesting.

MINOR COMMENTS:

  • Lines 106-108: In the sentence “One 106 of the vehicles, as the front vehicle, ran normally along the experiment route with a velocity of 60±5km/h. and a participant drove the following one.” there is a full stop after 60±5km/h which should be deleted.
  • Line 224: Please fix the text layout of line 224.
  • Line 342: In the sentence “In order to ensure the data were valid, we t carried out he tests on the effectiveness of emotional induction.” there is a typing error.
  • Lines 433-434: With reference to the sentence “Through the above processing and calculation, the change rates of the driving intention probability between the driver’s anger and natural state” I think there is something missing. Can you please reformulate it?
  • Lines 443-444-445: In the sentence “In the moderate or low Sd, the probability of keeping speed decreased significantly and the probability of the other two intentions increased observably when the Rdwas moderate or low.” there should be a space between Rd and
  • Lines 456-457: In the sentence “When the was moderate, the change was obviously only when the Rdwas high.” there should be a space between Rd and
  • Lines 461-462: Please check the sentence “It has generally accepted that fear emotion would increase the driver’s risk 461 perception [66], and the driver would prefer to choose the driving pattern with low risk [67].”. It should be “It is generally accepted” or “It has been generally accepted”.

Round 2

Reviewer 2 Report

The authors spent a much effort in revising their manuscript and seem to have greatly improved the manuscript. However, my considerations regarding some major methodological concerns could not be resolved completely.

Major concerns

1. Mainly, I still hold concerns regarding the absence of statistical test results in the revised version of the study.

Although the authors explanations in the response/cover letter greatly advanced my understanding of the model fitting procedure, the conclusiveness of the results lack statistical validity. I now understand the details of the model fitting procedure, however, my concerns about statistical statements such as “prediction accuracy of the proposed model for the deceleration, keeping speed, and acceleration intentions of the drivers reached a high level” (p.7, l.211f) still hold, as I do not understand how arbitrary accuracy values can be classified as either high or low. Regarding this issue, I would strongly recommend implementing and reporting statistical analysis to show that the obtained accuracy values are (1) higher than chance level and (2) higher than comparable computational models of driving intentions.

I hold similar concerns for statements in the discussion section, where the authors state, for example, that “drivers were more inclined to accelerate when they were in an angry state, i.e. preferred to seek a higher driving speed and a shorter following distance” (p.14, l.451f). Again, I would strongly recommend implementing and reporting statistical analysis to show that (1) the transition probabilities within a specific emotion are significantly different from one another and (2) changes in driving intentions for the specific emotions differ significantly from driving intentions without emotion (as reported in Study 1).

Reporting statistical analyses using significance tests (or at least quantifications) is important to obtain sound results on which interpretations can be based. A major step towards this would also be to include not only mean values, but also standard deviations, which the authors should provide for all figures.

2. Regarding my previous (minor) concern on the “time window for a single state”, which was set to 10 seconds (now p.4, l.135f), the authors responded that they “believe [sic] that a change in time window will not lead to a significant change in results”. Could the authors please provide empirical evidence for this claim by providing the reviewers with results for different time windows, e.g. 5 and 15 seconds? Although this means extensive reanalysis of the data, the lack of references in the literature calls for a careful evaluation of the time window, which is why I now consider this as a major concern.

Minor concerns

  1. The authors invest great effort in inducing and measuring emotions. However, the use of SAM ratings during driving seems to be highly impracticable. Could you please provide a short discussion on how emotional states could be reliably measured in real-world driving?
  2. Regarding my previous wish for clarification of the formal conflict between the statements that “if the probability of one driving intention increased, the transition probability of the other two driving intentions to it would increase. On the contrary, the transition probability of the other two intentions to it would reduce.” (p.12f, l.393ff). Although the provided example in the cover letter is clear, the authors should provide further clarification, as both sentences are almost identical, but end with either “increase” or “reduce”, which is paradox. If “in contrary” refers to the idea, that the probability of one driving intention now decreases, the authors should include this in the manuscript.
  3. If the concepts of demand satisfaction and risk degree are defined within in the framework of car-following theory, previous research should have used similar concepts. Please provide relevant links to the existing literature or include the theoretical considerations in the manuscript, as this seems to be highly relevant for the reader.

Reviewer 3 Report

I think the paper has been significantly improved and now warrants publication in IJERPH.
